# Modified Biopolymer Adsorbents for Column Treatment of Sulfate Species in Saline Aquifers

**DOI:** 10.3390/ma13102408

**Published:** 2020-05-23

**Authors:** Mostafa Solgi, Lope G. Tabil, Lee D. Wilson

**Affiliations:** 1Department of Chemistry, University of Saskatchewan, 110 Science Place, Saskatoon, Saskatchewan S7N 5C9, Canada; mos023@mail.usask.ca; 2Department of Chemical and Biological Engineering, University of Saskatchewan, 57 Campus Drive, Saskatoon, Saskatchewan S7N 5A9, Canada; lope.tabil@usask.ca

**Keywords:** fixed-bed column, adsorption, chitosan pellets, Thomas model, Yoon-Nelson model

## Abstract

In the present study, variable forms of pelletized chitosan adsorbents were prepared and their sulfate uptake properties in aqueous solution was studied in a fixed-bed column system. Unmodified chitosan pellets (CP), cross-linked chitosan pellets with glutaraldehyde (CL–CP), and calcium-doped forms of these pellets (Ca–CP, Ca–CL–CP) were prepared, where the removal efficiencies and breakthrough curves were studied. Dynamic adsorption experiments were conducted at pH 4.5 and 6.5 with a specific flow rate of 3 mL/min, fixed-bed height of 200 mm, and an initial sulfate concentration of 1000 mg/L. Breakthrough parameters demonstrated that Ca–CP had the best sulfate removal among the adsorbents, where the following adsorption parameters were obtained: breakthrough time (75 min), exhaust time (300 min), maximum sulfate adsorption capacity (q_max_; 46.6 mg/g), and sulfate removal (57%) at pH 4.5. Two well-known kinetic adsorption models, Thomas and Yoon-Nelson, were fitted to the experimental kinetic data to characterize the breakthrough curves. The fixed-bed column experimental results were well-fitted by both models and the maximum adsorption capacity (46.9 mg/g) obtained was for the Ca–CP adsorbent. A regeneration study over four adsorption-desorption cycles suggested that Ca–CP is a promising adsorbent for sulfate removal in a fixed-bed column system.

## 1. Introduction

Groundwater is the principal source of fresh water on Earth that represents an essential supply for domestic, agricultural, and industrial activities [1]. Saline groundwater is a major environmental concern for agricultural activities such as livestock farming. Various point sources of salinity arise from natural dissolution of minerals (carbonate, gypsum, sulfate-salts, etc.), atmospheric decomposition, and anthropogenic sources such as synthetic and organic fertilizer, municipal and industrial landfills, and mine tailings [2]. Among the various oxyanions, sulfate anion species in groundwater is problematic due to its higher solubility, as evidenced by reports of elevated concentration of sulfate in groundwater [3,4]. For instance, a published report by Feist et al. (2017) outlined information on the sulfate levels in well water of southern Saskatchewan, Canada [5]. Therein, the reported levels of sulfate ranged from 2000 to 6500 mg/L, where some samples of dugout water reached 20,000 mg/L. According to Health Canada guidelines for drinking water quality, the taste threshold recommended for sulfate in drinking water is 500 mg/L [6]. While sulfate is not considered a toxic contaminant when compared to heavy metals or fluoride, elevated sulfate can cause physiological effects such as diarrhea, dehydration, and changes in levels of methaemoglobin [7].

Various sulfate removal methods have been employed for treatment of wastewater and groundwater including chemical precipitation, electrocoagulation, biological sulfate reduction via sulfate reducing bacteria, ion exchange (IE) resins, and adsorption [8,9,10,11,12]. Among these techniques, adsorption-based processes have received considerable interest compared with other strategies due to their low operational cost and simplified technological design [13,14]. Choosing an appropriate adsorbent with high selectivity toward a specific type of contamination is a key factor in industrial water-treatment plants. Chitosan is the second most abundant biopolymer composed of β-linked glucosamine and N-acetyl-glucosamine units. The polar hydroxyl and amine functional groups of chitosan result in exceptional performance for the adsorption of cation and anion species [15]. Various forms of chitosan-based adsorbents have been developed to expand the field of application of chitosan and to address the controlled removal of contaminants from water and wastewater. Different types of chemical modification include cross-linking via glutaraldehyde or epichlorohydrin, formation of bio-composite pellets with biomass, and metal-imbibing methods have been applied to improve the mechanical properties of pristine chitosan, increase the surface area of adsorbents, and to utilize such adsorbents in acidic media [16]. Sowmya and Meenakshi (2013) studied the removal of nitrate and phosphate anions with protonated cross-linked chitosan beads (PCB) and carboxylated cross-linked chitosan beads (CCB) to study the effects of various operating parameters (e.g., pH, contact time, and adsorbent dosage). The maximum adsorption capacity (q_max_) for nitrate and phosphate by PCB at ambient temperature were reported as 113.1 and 58.5 mg/g, respectively, whereas CCB showed a lower q_max_ for nitrate (90.7 mg/g) and phosphate (16.6 mg/g) [17]. In another study by Liu et al. (2015), the application of zirconium immobilized cross-linked chitosan (Zr-CCS) for the removal of fluoride revealed that the adsorption process was spontaneous and endothermic with a q_max_ of 48.3 mg/g (fluoride) [18]. Jang and Lee conducted a study on phosphorus removal from aqueous media using chitosan/calcium-organically modified montmorillonite (chitosan/Ca-OMMT) beads in batch and fixed-bed column systems [19]. Their results covered a range of conditions (pH 3–11) with competitor anions (Cl^−^, NO_3_^−^, SO_4_^−^, and HCO_3_^−^) that revealed that chitosan/Ca-OMMT is a promising adsorbent for phosphate removal [19]. The Yoon-Nelson model was an appropriate theory for describing the adsorption process in a fixed-bed system, where the uptake of phosphate (q_max_ = 75.4 mg/g) by the chitosan/Ca-OMMT beads was higher than other reported chitosan bead systems. By comparison, examples of sulfate removal by conventional chitosan-based adsorbents are sparsely reported [20].

To the best of our knowledge, there is a limited number of reports on studies of sulfate adsorption in fixed-bed columns in the open literature [21]. The study reported herein for sulfate adsorption using a fixed-bed column with chitosan-based adsorbent represents a first example of its type. Considering the lack of data in the application of chitosan in fixed-bed columns for sulfate removal from aqueous solution, several objectives for this study are presented: (1) to prepare chitosan-based pellets as fixed-bed column materials; (2) to investigate the dynamic sulfate adsorption properties of the column bed system under a range of experimental conditions (bed height, pH, and influent levels); (3) to investigate kinetic models for analysis of kinetic adsorption results; and (4) to evaluate the sulfate uptake performance of the pelletized adsorbents over multiple adsorption-desorption cycles.

## 2. Materials and Methods

### 2.1. Chemicals

Low-molecular-weight chitosan (degree of deacetylation: 75–85% MW: 50,000−190,000 Da), CaCl_2_, BaCl_2_, NaCl (ACS grade) and KBr (IR Grade) were obtained from Sigma Aldrich Canada (Oakville, Ontario, Canada). Na_2_SO_4_, NaOH, glacial acetic acid and hydrochloric acid solution, were purchased from Fisher Chemical (Fisher Scientific, Hampton, NH, USA). Deionized water was used for the preparation of all standard solutions.

### 2.2. Experimental Methods

#### 2.2.1. Preparation of Adsorbents

Different types of pelletized adsorbents were prepared, as follows: chitosan pellets (CP), cross-linked chitosan pellets with glutaraldehyde (CL–CP), and calcium-doped forms of these pellets with and without cross-linking (Ca–CP, Ca–CL–CP). Chitosan pellets were prepared by dissolving 5 g chitosan powder in 15 mL of 0.2 M acetic acid, followed by an extrusion method using a glass syringe to make a pelletized adsorbent with an average diameter of 1.5 to 2 mm. The procedure of cross-linking was conducted at pH 5.6 by soaking the chitosan pellets in a glutaraldehyde (GA) solution for 48 h (CP/GA molar ratio was 1:1). After the cross-linking reaction, the pellets were washed extensively with Millipore water to obtain a neutral pH and to remove unreacted glutaraldehyde from the surface of pellets, followed by drying at room temperature. Calcium imbibing on the surface of CP and CL–CP was performed by soaking the pellets in calcium chloride solution (0.1 M) for 48 h.

#### 2.2.2. Characterization

Fourier transform infrared (FTIR) spectra were recorded using a Bio-RAD FTS-40 IR spectrophotometer (Bio-RAD Laboratories, Inc., Hercules, CA, USA) in reflectance mode to characterize the functional groups of the pellets. For preparation of samples, 6 mg of crushed pellets in powdered form were mixed with 60 mg of spectroscopic grade KBr. A total of 256 scans with a spectral resolution of 4 cm^−1^ were collected over a fixed wavenumber range (400–4000 cm^−1^), where the background spectrum of potassium bromide was subtracted. Thermogravimetric analysis (TGA) profiles of pellets were obtained using a TA Instruments Q50 TGA system (TA instruments, New Castle, DE, USA) to determine the thermal stability of adsorbents in terms of weight loss profiles and derivative thermal analysis (DTA). Samples were heated in open aluminum pans to a maximum temperature of 500 °C at a constant heating rate (5 °C/min) under a nitrogen atmosphere as the carrier gas.

#### 2.2.3. Dynamic Adsorption Tests

The fixed-bed adsorption experiments were performed in a glass column with a 300 mm height and 16 mm internal diameter. Initially, the pellets were soaked in Millipore water for 24 h to reach maximum swelling and to remove air bubbles. After that, pellets were packed in the column by the slurry method. The height of the glass beads was fixed (30 mm) at the bottom and top parts of the column for better flow rate. In all experiments, the bed height was 200 mm, and the pH was either 4.5 or 6.5. The feed solution containing sulfate solution (1000 mg/L) was pumped upward with constant flow rate (3 mL/min) by a peristaltic pump. Before the adsorption experiment, 200 mL Millipore water at pH 4.5 or 6.5 was passed through the column to wash the pellets and adjust the flow rate. The effluent was collected at different time intervals to determine the residual concentration of sulfate, where it was measured via a ultraviolet–visible (UV–vis)-based turbidity method that employs BaCl_2_ [22]. When the sulfate ion concentration of the effluent converged with the concentration to that of the influent, the pellets were considered as completely saturated and the adsorption process was stopped. Figure 1 shows the fixed-bed column set up and bed packed with pellets.

## 3. Result and Discussion

As outlined above, chitosan adsorbents were prepared in a pelletized form where synthetic modification was carried out based on well-established synthetic methods developed in-house and by other groups [13,15,16,17,18]. In this fashion, the materials characterization herein was focused on the use of TGA and IR spectroscopic methods, as outlined below, along with characterization of the sulfate adsorption properties under kinetic conditions using a fixed-bed column setup. The results for the characterization of the pelletized materials and their adsorption properties are outlined in the sections that follow.

### 3.1. Functional Groups on Adsorbents

FTIR spectroscopy has been employed as a useful characterization technique to evaluate the type of functional groups present in the adsorbent material. The FTIR spectra of CP, Ca–CP, CL–CP, and Ca–CL–CP are presented in Figure 2 without normalization.

The peaks occurred in 3000–3500 cm^−1^ region correspond to stretching vibrations of −OH and −NH_2_ groups [23]. The bands at 2857 and 2879 cm^−1^ relate to the presence of C–H stretching vibrations. The signature at 1660 cm^−1^ is ascribed to presence of occluded acetic acid within the pellets according to the mode of preparation [16]. The IR band near 1554 cm^−1^ is assigned to the N–H bonds. The bands in the range of 1100–1300 cm^−1^ correspond to C–O–H, C–O–C, and C–N–H. Moreover, the band at 1080 and 890 cm^−1^ relates to C–O and C–H stretching, respectively [24]. In Figure 2b, it was observed that the stretching vibration of O–H and N–H at 3442 cm^−1^ in CL–CP shifted to 3431 cm^−1^ in Ca–CL–CP. Furthermore, the band at 1578 cm^−1^, which is related to N–H bending, shifted to 1597 cm^−1^. By comparing FTIR spectra of CP (Figure 2a) and CL–CP (Figure 2b), O–H bending vibration at 1379 cm^−1^ and C–O bending vibration at 1263 cm^−1^ disappeared completely. Moreover, the intensity of the N–H band near 3500 cm^−1^ in Ca–CP and Ca–CL–CP was decreased which shows the attachment of metal ions to amine groups on the surface of chitosan-based adsorbents. Vafakish and Wilson reported parallel changes in the IR spectra upon the adsorption of Cu(II) by an aniline-grafted chitosan adsorbent [13].

### 3.2. Thermal Stability

The weight loss profiles and derivative thermal analysis (DTA) for CP, Ca–CP, CL–CP, and Ca–CL–CP are presented in Figure 3a,b.

Pellets without cross-linking showed two stages of thermal degradation. The first thermal event between 30 °C and 150 °C was related to the loss of acetic acid and water content present in the pellets. For chitosan pellets (CP), decomposition occurred between 250 °C and 350 °C with a sharp thermal event near 300 °C, whereas Ca–CP decomposition occurred from 220–330 °C. For pellets with cross-linking agent, three stages of thermal decomposition were observed. The first stage of thermal degradation occurred between 30 °C to 150 °C and was attributed to loss of acetic acid and moisture. For cross-linked pellets (CL–CP), the decomposition temperature was shifted to a lower temperature range (210 to 340 °C). Besides, a thermal event at 430 °C for CL–CP and Ca–CL–CP was observed that relates to the decomposition of the glutaraldehyde moiety, in agreement with another related study [17]. For both calcium imbibed pellets (Ca–CP and Ca–CL–CP), the thermal event was observed at lower temperature, as compared to pellets without calcium modification. For Ca–CP and Ca–CL–CP, decomposition occurred between 220–330 °C. The observed temperature shift for the calcium imbibed samples relates to the coordination of metal-ion, leading to a reduction of the intra- and inter-chain hydrogen bonding effects of chitosan [25]. The TGA curves reveal that the chitosan-based adsorbents reported in this study are stable up to 210 °C which is comparable to the results reported by Tavares et al. [26]. As well, the shifts in thermal stability (cf. Figure 3) are consistent with the trends in chemical modification due to the trend in IR results in Figure 2 for calcium imbibing and glutaraldehyde cross-linking, as reported elsewhere [25].

### 3.3. Breakthrough Parameters for the Sulfate Adsorption Isotherms

By analyzing the breakthrough curves in a dynamic adsorption process, the performance of a fixed-bed adsorption column can be evaluated. The practical feasibility and economics of an adsorption process can be evaluated through analysis of the breakthrough time (t_b_), exhaustion time (t_s_) and shape of breakthrough curve, where these parameters are important considerations in the design of a fixed-bed adsorption column. The breakthrough curve is a plot of normalized concentration (C_t_/C_0_) as a function of time or volume of effluent for a given bed height, where C_t_ is the effluent concentration and C_o_ is the concentration of influent. The breakthrough time (t_b_) is the point where effluent concentration (C_t_) from the column is ca. 5% of the initial concentration (C_t_/C_0_ = 0.05) and the exhaustion time (t_s_) is the point where the effluent concentration reached 95% of the initial concentration (C_t_/C_0_ = 0.95). Several key parameters can be calculated from an analysis of the breakthrough curves, as follows:

The total amount of adsorbed $SO_{4}^{2-}$ anions (q_total_) and maximum adsorption capacity (q_max_) can be evaluated with Equations (1) and (2), respectively [27].
(1)$$q_{\mathrm{total}}=QC_{o}A=QC_{0}\int_{t=0}^{t=t_{\mathrm{total}}}\left(1-\frac{C_{t}}{C_{0}}\right)dt$$
(2)$$q_{\max}=\frac{q_{\mathrm{total}}}{M}$$
where t_total_ represents the total flow time, C_0_ is the initial sulfate concentration (mg/L), C_t_ (mg/L) is the sulfate concentration after adsorption for time (t), M is the mass of adsorbent, Q is the flow rate (L/h) and A is the area under breakthrough curve.

The total amount of sulfate ions (q_total_) flowing into the fixed-bed column can be obtained by Equation (3):(3)$$m_{\mathrm{total}}=C_{0}Qt_{\mathrm{total}}$$

The removal (%) of sulfate ions can be calculated according to Equation (4).
(4)$$\mathrm{Removal}\left(\%\right)=\frac{q_{\mathrm{total}}}{m_{\mathrm{total}}}\times 100$$

The mass transfer zone (MTZ) shows the efficiency of a specific adsorbent as the length of the adsorption zone in the column where the adsorption of solutes takes place. MTZ is calculated by Equation (5) [28].
(5)$$\mathrm{MTZ}=Z\times \left(1-\frac{t_{b}}{t_{s}}\right)$$
where, Z is total depth of the adsorbent in the column (mm).

The total effluent volume (V_s_) and breakthrough volume (V_b_) can be obtained by Equations (6) and (7), respectively.
(6)$$V_{s}=Q\times t_{s}$$
(7)$$V_{b}=Q\times t_{b}$$

The breakthrough curves at pH 4.5 and pH 6.5 are shown in Figure 4.

The conditions of these adsorption experiments were performed at fixed initial sulfate concentration (1000 mg/L), flow rate (3 mL/min), and a constant bed height (200 mm). In Figure 4, the breakthrough curves for all adsorbents was shifted from right to left upon increase of the pH from 4.5 to 6.5. This trend indicates a reduction in the uptake of sulfate ions from pH 4.5 to 6.5, in parallel agreement with a decrease in the number of protonated amine groups and other active sites on the adsorbent surface. This trend is evident for each specific adsorbent in terms of maximum adsorption capacity, breakthrough time, and exhaustion time. The value of q_total_ at pH 6.5 for each adsorbent decreased when compared to those at pH 4.5 (Table 1).

By analyzing the level of removal (%), the adsorbents at pH 4.5 and 6.5 could remove 50% or more sulfate ions for each cycle of treatment in the fixed-bed column (Table 1). These results suggest that the adsorbents reported herein have potential utility for sulfate removal. However, the best performance was noted for Ca–CP at pH 4.5 with a maximum adsorption capacity 46.6 mg/g and a removal level of 57%. A comparison of the breakthrough parameters among the various adsorbents at pH 4.5 and 6.5, the breakthrough and exhaustion times for Ca–CP is highest among the various adsorbents. Thus, the Ca–CP system can adsorb more sulfate ions versus the other adsorbents. In terms of breakthrough time (t_b_) and exhaustion time (t_s_) for CP, CL-CP, and Ca–CL–CP, no significant difference was observed.

### 3.4. Kinetic Models for Sulfate Adsorption in the Continuous Fixed-Bed Column

#### 3.4.1. Thomas Model

The Thomas model provides a general analytical description to predict the behaviour of dynamic adsorption processes. The Thomas model is defined based on the Langmuir kinetics of adsorption and assumes that the system has a constant flow rate where there is no axial dispersion in continuous adsorption, and second order reversible kinetics applies to the rate of the driving force. The main limitation of the Thomas model is that it considers that sorption is not dominated by a chemical process, but rather by mass transfer at the interface. The linear form of Thomas model is shown by Equation (8), where the model constants (k_th_ and q_max_) were obtained [29].
(8)$$\ln\left[\frac{C_{0}}{C_{t}}-1\right]=\frac{k_{\mathrm{th}}q_{\max}M}{Q}-k_{\mathrm{th}}C_{0}t$$
where k_th_ (L mg^−1^ h^−1^) is the Thomas model constant, q_max_ (mg/g) is adsorption capacity, M (g) is the quantity of sorbent, C_0_ (mg L^−1^) is the influent concentration, C_t_ (mg L^−1^) is the effluent concentration, and t (h) is time.

The linearized form of Thomas model for the various adsorbents at pH 4.5 and pH 6.5 are presented in Figure 5.

The values of the model constant (k_th_) and theoretical adsorption capacity (q_max_) were calculated using a linear regression analysis according to Equation (8), as summarized in Table 2.

At pH 4.5, the highest adsorption capacity obtained was obtained for Ca–CP with 46.9 mg/g and the lowest adsorption capacity was that of CL–CP (q_max_ = 29.6 mg/g). These results showed that the number of calcium ions attached to the adsorbent surface of Ca–CP was higher than that for Ca–CL–CP which results in a greater adsorption capacity. Also, the presence of Ca^2+^ on the surface of adsorbents resulted in a higher adsorption capacity for calcium imbibed pellets when compared to the other adsorbents (CP, and CL–CP). The values of k_th_ at pH 4.5 were obtained in the range of 0.0016–0.0018 L mg^−1^ h^−1^. The highest value of k_th_ was obtained for Ca–CL–CP, while the lowest value was noted for Ca–CP. Similar trends for the values of q_max_ and k_th_ were obtained at pH 6.5, where the best performance was observed by Ca–CP (q_max_ = 30.3 mg/g) and the lowest value was noted for CL–CP(q_max_ = 22.0 mg/g). The coefficient of correlation (R^2^) values at pH 4.5 and 6.5 were nearly unity (R^2^ ≈ 1), suggesting that this model resulted in a good fit to the experimental data.

#### 3.4.2. Yoon–Nelson Model

In the Yoon–Nelson model, there is no input data on the adsorbent physical properties in the fixed-bed column. This model relates the decrease in the probability of adsorption for each sorbate molecule is proportional to the probability of sorbate adsorption and the probability of sorbate breakthrough from the sorbent material [30]. The linear form of the Yoon–Nelson model is presented in Equation (9), where the model constants k_YN_ and *τ* are obtained.
(9)$$\ln\left[\frac{C_{t}}{C_{0}-C_{t}}\right]=k_{\mathrm{YN}}t-\tau k_{\mathrm{YN}}$$
where k_YN_ (h^−1^) is the rate velocity constant and *τ* (h) is time needed for 50% adsorbate breakthrough. This model has the ability to predict the time for 50% adsorbate breakthrough.

The linearized form of the Yoon–Nelson model for the various adsorbents is presented in Figure 6.

The model constants k_YN_ and τ values were calculated using linear regression analysis according to Equation (9) and are summarized in Table 3. The values of coefficient of determination (R^2^), ranged from 0.98 to 0.99, at pH 4.5 and 6.5, where this model provided an excellent fit to the experimental results. The k_YN_ is a rate constant that explains the diffusion characteristics of the mass transfer zone (h^−1^). At pH 4.5, Ca–CP showed the best adsorption properties among the adsorbents, where k_YN_ is the lowest value with 1.60 h^−1^. This result indicates that mass transfer by diffusion inside the column packed by Ca–CP was better than other adsorbents. Whereas, the highest value of k_YN_ is 2.29 h^−1^ for the column with the CL–CP adsorbent. This indicates that lower diffusion was obtained for the column packed with CL–CP when compared to the other adsorbents. In a dynamic adsorption process in a fixed-bed column, a high τ-value is more desirable because it indicates that more adsorbate can be bound by a specific adsorbent. Among the four types of adsorbents tested herein at pH 4.5 and 6.5, Ca–CP has the highest τ value. These results are in accordance with the breakthrough curve results which reveal that the maximum adsorption capacity, breakthrough time (t_b_) and exhaustion time (t_b_) for Ca–CP were the highest among the adsorbents studied herein.

### 3.5. Effect of Operating Conditions on Column Efficiency

In a dynamic adsorption process, different operating parameters including pH, flow rate, initial concentration of adsorbate, bed height, and many others can affect the performance of the column. It is important for industrial units to use these optimized parameters to reach the highest performance of a fixed-bed reactor at an optimum operational cost. According to the experimental data presented in Table 1, along with the kinetic studies for the various adsorbent materials analyzed by the Thomas and Yoon-Nelson models, Ca–CP was chosen for further study. The results for Ca–CP at pH of 4.5 yielded the best adsorption, illustrating the need to further investigate the effects of variable operating parameters on the column performance.

#### 3.5.1. Bed Height

It is known that the efficiency of a fixed-bed column system can be influenced by changing the column bed height. The importance of this parameter relates to the amount of adsorbent used in the column design that affects the quality of effluent [31]. Figure 7a shows the effect of different bed heights (200 mm and 300 mm) on the breakthrough curves of sulfate adsorption by Ca–CP pellets. As can be seen from Figure 7a, lower bed height led to shorter breakthrough time or steeper breakthrough curves. The results showed that the adsorption capacity increased from 46.6 to 63.8 mg/g with a rise in the bed height from 200 to 300 mm (Table 4).

An increased breakthrough time (140 min) was obtained with greater bed height. Likewise, an increase in the bed height from 200 mm to 300 mm yielded an increase in the exhaustion time from 225 to 420 min. Nguyen et al. [27] reported similar observations for phosphate adsorption by Zr (IV)-loaded okara (ZLO) in a dynamic adsorption study. This effect can be accounted for by two experimental trends: i) an increased bed height at a fixed feed concentration provides a larger surface area with more active sites on the surface of Ca–CP pellets that yield greater adsorption capacity [32], and ii) the role of diffusion phenomena predominates in mass transfer and reduced axial dispersion as the bed depth is increased. As a result, sulfate ions have more time to diffuse into the pellets [33].

#### 3.5.2. Initial Sulfate Concentration

The effect of the initial sulfate concentration (1000 mg/L and 2000 mg/L) on the performance of the column at constant flow rate (3 mL/min) and bed height (200 mm) is presented in Figure 7b. The results indicate that sulfate ion levels in the influent can affect the breakthrough curve to a significant extent. It is obvious from Figure 7b that the breakthrough time and exhaustion time decreased remarkably at high initial sulfate concentration. When the sulfate level in the influent increased from 1000 mg/L to 2000 mg/L, the maximum adsorption capacity increased from 46.6 mg/g to 49.1 mg/g (cf. Table 4). Similarly, it can be seen from Table 4 that the level of sulfate removal (%) significantly increased from 57% to 85.7% as the influent concentration doubled. The concentration gradient of the adsorbate in solution acts as a driving force for mass transfer, where a higher initial sulfate level leads to greater mass transfer of adsorbate to the Ca–CP adsorbent surface. In turn, this phenomenon leads to higher adsorption capacity and sulfate removal [34].

#### 3.5.3. Flow Rate

To investigate the effect of flow rate on the performance of a fixed-bed column, two different flow rates (3 and 5 mL/min) at constant feed concentration (1000 mg/L) and bed height (200 mm) were examined. The results demonstrate that the breakthrough curve shifted to the left with an increase in the maximum adsorption capacity (Figure 7c). On the other hand, as the flow rate increased from 3 to 5 mL/min, the breakthrough time and the exhaustion time decreased (Figure 7c). As well, the q_max_ increased from 46.6 to 50.5 mg/g for an increase in flow rate from 3 to 5 mL/min (Table 4). Furthermore, the breakthrough time decreased from 75 to 45 min as the flow rate increased from 3 to 5 mL/min. Also, by increasing the flow rate, the saturation of packed Ca–CP was reached in a shorter time. The removal (%) of sulfate was not affected measurably by increasing the flow rate, according to ca. 5% decrease (Table 4). Basu et al. reported that more metal ions come in contact with the adsorbents at higher flow rate causing higher metal loading and an increased uptake. As the flow rate increased, the contact time between adsorbate and the surface of adsorbent decreased and mobile sulfate ions pass through the column faster, resulting in a shortened breakthrough and exhaustion time [29].

### 3.6. Dynamic Modeling of Sulfate Adsorption in Ca–CP Pellets Using the Thomas Model

#### 3.6.1. Effect of Bed Height

The linear form of the Thomas model, Equation (8), was used to obtain the model parameters (k_th_ and q_max_) at bed heights of 200 and 300 mm (cf. Appendix A). The value of k_th_ decreased from 0.0016 to 0.0014 L mg^−1^ h^−1^ when the bed height increased from 200 mm to 300 mm. The values obtained by the Thomas model for q_max_ are very close to experimental values obtained for the maximum adsorption capacity.

#### 3.6.2. Effect of Initial Sulfate Concentration

The linear form of the Thomas model at different initial sulfate concentration is presented in Appendix A. In Table 5, the maximum adsorption capacity (q_max_) increased from 46.9 to 56.7 mg/g as the initial sulfate concentration in the influent doubled (1000 to 2000 mg/L) at a fixed-bed height. As indicated above, the higher sulfate influent concentration leads to a higher concentration gradient that results in an increased column performance [27].

#### 3.6.3. Effect of Flow Rate

During the adsorption process, the contact time between the adsorbent and adsorbate is key parameter because the diffusion of ions from solution to the surface of adsorbent is time dependent. To determine the effect of flow rate on the adsorption of sulfate onto Ca–CP pellets, kinetic experiments were conducted at variable flow rate of 3 and 5 mL/min (cf. Appendix A). Basu et al. reported that the adsorbent undergoes saturation more rapidly as the flow rate increases and the adsorption process occurs over a shorter time interval [29]. This relates to the fact that as the flow rate increases, there is non-optimal contact between ions and the active sites on the adsorbent surface. The results of the current study relate to the effect of flow rate on the adsorption process, and show parallel agreement with other related adsorbent systems [29]. In conclusion, for different operating parameters, the value of coefficient of determination (R^2^) varied from 0.92 to 98, indicating that the Thomas model accounts for most of the deviation of the experimental data (cf. Table 5).

### 3.7. Desorption Study

For the study of desorption, an exhausted calcium-doped (Ca–CP) material was considered herein. After the adsorption step in cycle 1, 200 mL of Millipore water was passed through the column in order to desorb sulfate ions from the biopolymer pellet surface. The regeneration agent for desorption of sulfate was 0.5 M NaCl (*aq*), where the bed height was constant at 200 mm. The NaCl solution with a flowrate of 3 mL/min in the upward direction was passed through the column. By analyzing the sulfate concentration in the effluent, it was determined that the total time for desorption was 45 min. After the desorption process by NaCl, 200 mL of Millipore water was used for washing the residual NaCl from the adsorbent surface, where this procedure was repeated for cycles 2 to 4.

The breakthrough curves for cycles 1 to 4 and the related parameters for this system are presented in Table 6 and Appendix A.

In Appendix A, there is no significant difference between the position of breakthrough curves for cycles 1 and 2, where the breakthrough time (75 min) and exhaustion time (300 min) are in close agreement (cf. Table 6). The difference in adsorption capacity and removal (%) of sulfate between cycles 1 to 2 are negligible, and reveal that the adsorption efficiency of Ca–CP over the first two cycles is comparable. On the other hand, there is a significant shift in the position of breakthrough curves from right to left for cycles 3 and 4. According to Table 6, the breakthrough time (45 min) and exhaustion time (250 min) in cycle 3 decreased significantly. Also, the adsorption capacity from cycle 2 to 3 decreased from 46.3 to 36.6 mg/g. However, a comparison of the adsorption capacity of sulfate ions for pellets used in cycle 4 compared to pristine pellets in cycle 1 reveal a decrease of nearly 50%. The decrease in the efficiency of adsorbent in cycles 3 and 4 relate to the leaching of adsorbed calcium ions from the adsorbent during the NaCl desorption process [25]. Based on this approach, the prepared Ca–CP in each cycle can typically remove more than 50% of sulfate species in the influent at pH 4.5.

Table 7 shows a comparison of maximum sulfate uptake by Ca–CP and other adsorbents at different operating parameters. In Table 7, the q_max_ value for chitin shrimp shell is 156 mg/g, where its greater values of q_max_ relative to Ca–CP (63.8 mg/g) relates to the higher surface area of the chitin material. In the former study, chitin was used in its powder form, where a greater surface accessible area for such particle morphololgy is anticipated, as compared to materials in their pelletized form, as reported herein. While, powder-based adsorbents have greater accessible surface area than other adsorbentmorphologies such as beads and pellets, there are known practical limitations with the use of powders. This includes limitations on the use of high operational pressures, high cleaning/regeneration cost of the exhausted column, and requirements of adsorbent packing of fresh column beds. The facile Ca–CP pellet preparation, biodegradability, practical utility of pellets for fixed-bed columns, and the relatively low preparation cost of Ca–CP pellets reveal their practical utility as an adsorbent for removing SO_4_^2−^ from groundwater at typical conditions.

## 4. Conclusions

In this study, we report a first example of chitosan-based adsorbents in a pelletized form that were used to for the removal of sulfate anions from laboratory prepared samples under dynamic column conditions. The adsorbents include pure chitosan pellets (CP), modified chitosan pellets with calcium (Ca–CP), cross-linked chitosan pellets (CL–CP) with glutaraldehyde, and CL–CPwith calcium imbibing (Ca–CL–CP). The study reveals that all adsorbents had variable sulfate uptake and efficiency at pH 4.5 or pH 6.5. Among the various adsorbents, Ca–CP showed the highest maximum adsorption capacity of 46.6 mg/g at pH 4.5 and 29.0 mg/g at pH 6.5. The greater efficiency of sulfate uptake at pH 4.5 relates to the availability and protonation of amine sites of chitosan that concur with the role of ion-exhange effects. The Thomas and Yoon–Nelson kinetic adsorption models provided favourable best-fit results to the sulfate uptake onto the Ca–CP adsorbent. The favourable uptake of Ca–CP among the various adsorbents led to its further investigation, where the role of operating parameters are summarized, as follows:Increased bed height resulted in greater adsorption capacity and sulfate removal;Greater flow rate (3 vs. 5 mL/min) led to increased maximum adsorption capacity (q_max_) but decreased sulfate removal (%), where premature sulfate breakthrough resulted at higher flow rate under such dynamic adsorption conditions;Greater intial sulfate concentration from 1000 to 2000 mg/L enhanced the driving force for mass transfer, that contributed to an increased maximum adsorption capacity;Sulfate adsorption onto Ca–CP adsorbent at variable operating parameters was well represented by the Thomas model; andThe desorption experiments with 0.5 M NaCl showed good performance of Ca–CP over the first 2 cycles, whereas a decrease was evident for further cycles.

## Figures and Tables

**Figure 1 materials-13-02408-f001:**
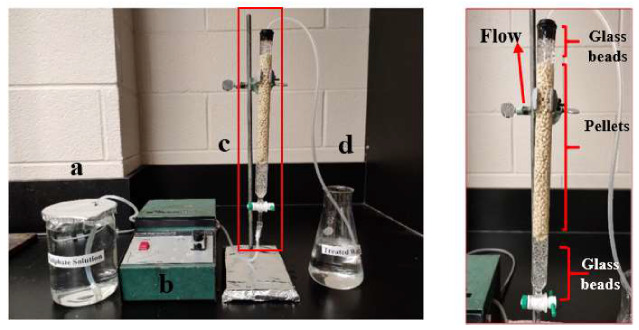
Fixed-bed column reactor packed with chitosan pellets (Ca-CP): (**a**) influent solution, (**b**) peristaltic pump, (**c**) column, and (**d**) influent. The reader is referred to the online version of this article for color images of all graphic files in this document.

**Figure 2 materials-13-02408-f002:**
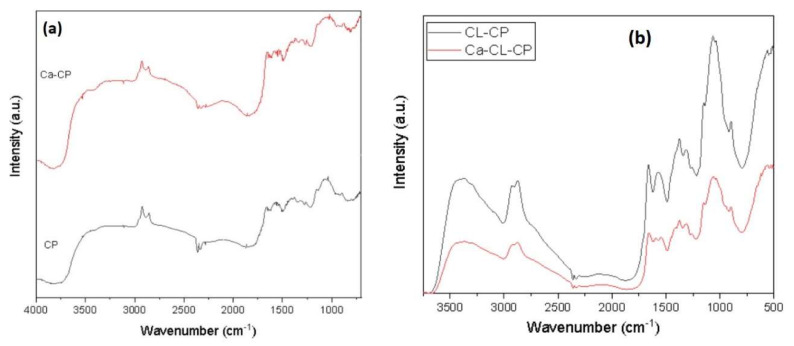
Fourier transform infrared (FTIR) spectra for pellets. (**a**) CP and Ca–CP, (**b**) CL–CP and Ca–CP.

**Figure 3 materials-13-02408-f003:**
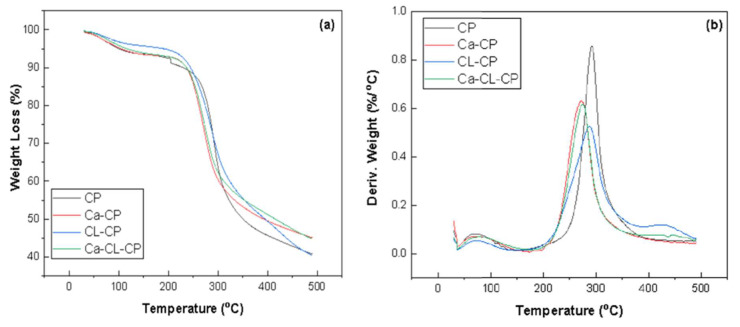
(**a**) Weight loss (%) and (**b**) differential thermal analysis (DTA) plots for CP, Ca–CP, CL–CP, and Ca–CL–CP.

**Figure 4 materials-13-02408-f004:**
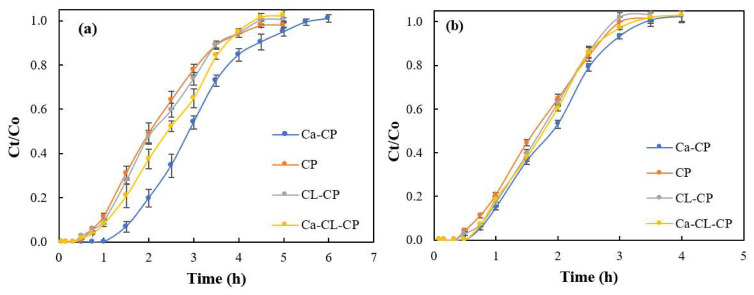
Breakthrough curves for the adsorption of sulfate by the various adsorbent systems: (**a**) pH 4.5, and (**b**) pH 6.5 (initial sulfate concentration = 1000 mg/L, 25 °C, flow rate of 3 mL/min, and bed height = 200 mm).

**Figure 5 materials-13-02408-f005:**
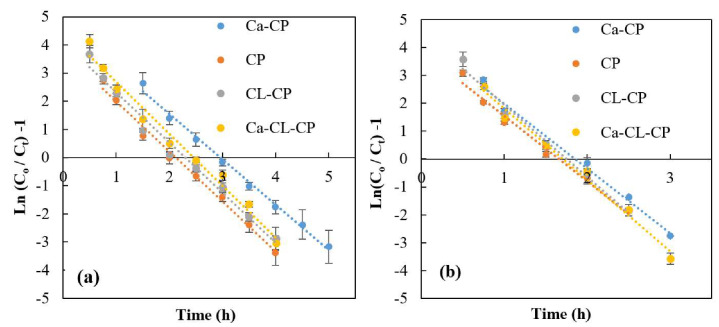
Linear plots of Thomas model for various adsorbents at 25 °C: (**a**) pH 4.5, and (**b**) pH 6.5, at an initial sulfate concentration = 1000 mg/L, flow rate of 3 mL/min, and bed depth of 200 mm.

**Figure 6 materials-13-02408-f006:**
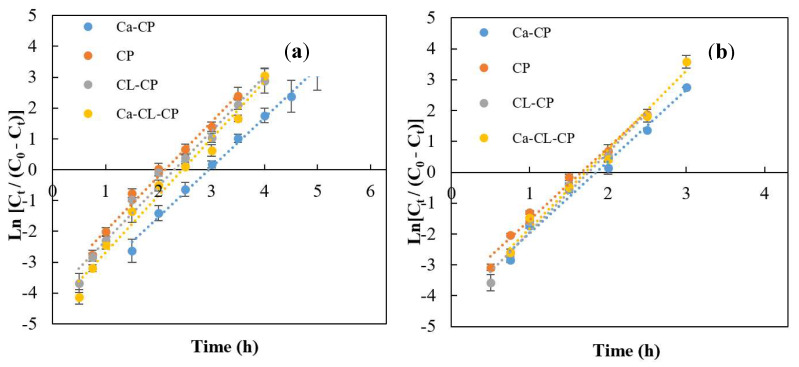
Linear plots of Yoon–Nelson model for various adsorbents at 25 °C: (**a**) pH 4.5, and (**b**) pH 6.5 at a fixed initial sulfate concentration (1000 mg/L), flow rate (3 mL/min), and a fixed-bed depth of 200 mm.

**Figure 7 materials-13-02408-f007:**
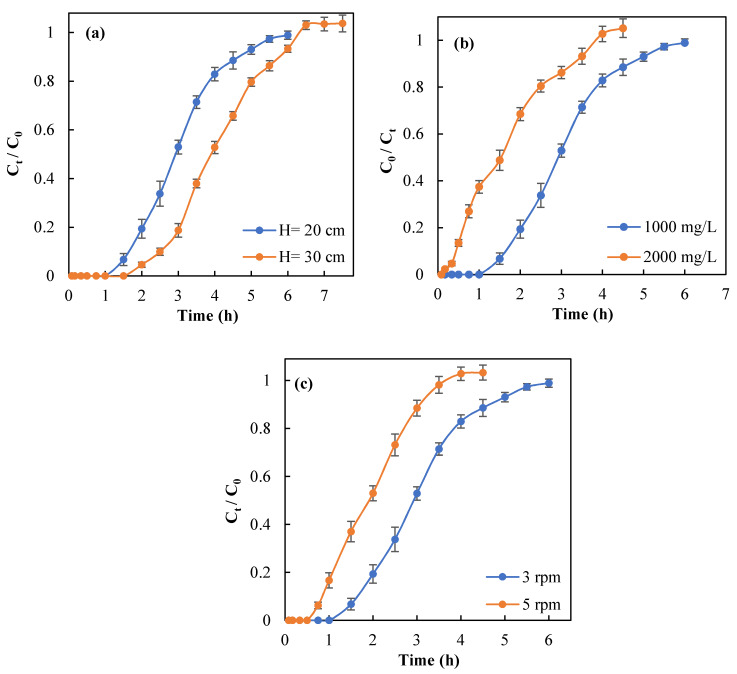
Breakthrough curves for sulfate adsorption onto the Ca–CP adsorbent at different operating conditions: (**a**) bed height, (**b**) initial sulfate concentration, and (**c**) flow rate.

**Table 1 materials-13-02408-t001:** Breakthrough curve parameters for variable adsorbents for the adsorption of $SO_{4}^{2-}$.

Adsorbent	M (g)	pH	t_b_ (min)	t_s_ (min)	R (%)	q_total_ (mg)	q_max_ (mg/g)	MTZ (mm)	V_b_ (mL)	V_s_ (mL)
Ca–CP	11	4.5	75	300	57.00	513.0	46.6	150	225	900
CP	14	4.5	42	250	50.25	361.8	32.1	160.6	126	750
CL–CP	11	4.5	45	250	53.40	384.5	27.5	160.4	135	750
CL–Ca–CP	14	4.5	50	250	56.00	420.5	30.0	160	150	750
Ca–CP	11	6.5	45	180	50.50	318.6	29.0	150	135	540
CP	14	6.5	32	160	50.00	270.0	24.5	160	96	480
CL–CP	11	6.5	35	170	54.30	293.4	21.0	150.9	105	510
CL–Ca–CP	14	6.5	35	170	55.30	298.8	21.3	150.9	105	510

**Table 2 materials-13-02408-t002:** Thomas model constants for different adsorbents.

Adsorbent	pH	Thomas Model Constant k_th_ × 10^3^ (L mg^−1^ h^−1^)	Adsorption Capacity q_max_ (mg/g)	R^2^
Ca–CP	6.5	2.3	30.3	0.98
Ca–CP	4.5	1.6	46.9	0.99
CP	6.5	2.3	27.2	0.98
CP	4.5	1.7	34.7	0.99
CL–CP	6.5	2.5	22.0	0.98
CL–CP	4.5	1.7	29.6	0.98
Ca–CL–CP	6.5	2.5	22.5	0.98
Ca–CL–CP	4.5	1.8	31.6	0.98

**Table 3 materials-13-02408-t003:** Yoon–Nelson model constants for different adsorbents at 25 °C and variable pH.

Adsorbent	pH	Yoon-Nelson Constant k_YN_ (h^−1^)	τ (h)	R^2^
Ca–CP	6.5	2.31	1.85	0.98
Ca–CP	4.5	1.60	2.95	0.99
CP	6.5	2.34	1.66	0.98
CP	4.5	2.12	1.77	0.98
CL–CP	6.5	2.85	1.75	0.98
CL–CP	4.5	2.29	1.78	0.98
Ca–CL–CP	6.5	2.55	1.71	0.98
Ca–CL–CP	4.5	1.84	2.46	0.98

**Table 4 materials-13-02408-t004:** Breakthrough curve parameters for sulfate uptake by Ca–CP at variable conditions.

Flow Rate [Q] (mL/min)	Bed Height [Z] (mm)	Feed Concentration [C_o_] (mg/L)	Breakthrough Time [t_b_] (min)	Exhaustion Time [t_s_] (min)	Adsorption Capacity [q_max_] (mg/g)	Removal [R] (%)
3	200	1000	75	300	46.6	57.00
3	200	2000	22	182	49.1	85.71
3	300	1000	140	370	63.8	78
5	200	1000	45	210	50.5	52.85

**Table 5 materials-13-02408-t005:** Thomas model constants for adsorption of sulfate onto Ca–CP.

Flow Rate (mL/min)	Bed Height (mm)	Feed Concentration (mg/L)	k_th_ × 10^3^ (L mg^−1^ h^−1^)	Adsorption Capacity (mg/g)	Coefficient of Correlation (R^2^)
3	200	1000	1.60	46.9	0.98
3	200	2000	0.86	56.7	0.92
3	300	1000	1.42	65.4	0.99
5	200	1000	1.97	52.7	0.97

**Table 6 materials-13-02408-t006:** Breakthrough curves parameters of sulfate adsorption on Ca–CP for cycles 1 to 4.

Cycle	Breakthrough Time (t_b_)	Exhaustion Time (t_s_)	Removal (%)	Adsorbed Sulfate q_total_ (mg)	Adsorption Capacity (mg/g)
**1**	75	300	57.0	513.0	46.6
**2**	75	300	56.5	509.0	46.3
**3**	45	250	56.2	425.0	36.6
**4**	20	150	55.0	268.2	24.4

**Table 7 materials-13-02408-t007:** Comparison of maximum adsorption capacity of Ca–CP with various sulfate adsorbents.

Adsorbent	Adsorption Capacity (mg/g)	Feed Concentration (mg/L)	Other Conditions	Ref.
Zirconium oxide-modified pomelo peel biochar	35.2	300	pH 5, 25 °C	[12]
Surfactant-modified palygorskite	3.28	130	pH 4, 35 °C	[35]
Organo-nano-clay modified with cetyltrimethylammonium bromide (CTAB)	38	500	pH 7, 40 °C	[36]
ZnCl_2_ activated coir pitch carbon	49	40	pH 4, 35 °C	[37]
γ-Al_2_O_3_	8.5	------	pH 5.7	[38]
CC/QAC	526	------	pH 5	[39]
PG-Peat	189.5	1835	pH 2.4, 22 °C	[11]
Modified rice straw	74.8	500	pH 6.4, 25 °C	[40]
Desilicated fly ash	147.1	------	35 °C	[41]
Polypyrolle-grafted granular activated carbon	44.7	250	pH 7, 20 °C	[42]
Ba-modified blast-furnace-slag-geopolymer	119	865	pH 7–8	[43]
Poly(*m*-Phenylendiamine)	109	109	pH < 3	[44]
Chitin	156	2325	pH 4.5	[20]
**Calcium-Chitosan pellet** (Ca–CP)	63.8	1000	pH 4.5	**This study**

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
