# Peer review of "Modified Biopolymer Adsorbents for Column Treatment of Sulfate Species in Saline Aquifers"

_materials, 2020, doi:10.3390/ma13102408_

Round 1

Reviewer 1 Report

The manuscript "Modified Biopolymer Adsorbents for Column Treatment of Sulfate Species in Saline Aquifers" (Manuscript ID: materials-808176) present the results of study conducted in order to test efficiency of unmodified chitosan pellets, cross-linked chitosan pellets with glutaraldehyde and calcium-doped forms of chitosan pellets in sulfate removal.

The efficiency of tested pellets were determinated by bed columns under various bed height, pH, and influent levels. Obtained results were processed and expressed by Thomas and Yoon-Nelson kinetic adsorption models.

Characterisation of used pellets was determined by FTIR.

This manuscript is well written, all obtained results are clearly presented.

The research design could be improved by additional adsorption tests i.e. by batch sorption experiments and determination (at least) Langmuir and Freundlich Isotherm Studies.

Additional structural analysis (thermogravimetric analysis, nuclear magnetic resonance and scanning electron microscopy) of synthesized pellets would also improve the quality of this manuscript.

If the authors can not provide additional informations related to structural analysis of tested adsorbens, the manuscript can be accepted in current form.

Author Response

Response to Reviewer Comments on MS ID: materials-808176

Reviewer #1

The manuscript "Modified Biopolymer Adsorbents for Column Treatment of Sulfate Species in Saline Aquifers" (Manuscript ID: materials-808176) present the results of study conducted in order to test efficiency of unmodified chitosan pellets, cross-linked chitosan pellets with glutaraldehyde and calcium-doped forms of chitosan pellets in sulfate removal.

The efficiency of tested pellets were determinated by bed columns under various bed height, pH, and influent levels. Obtained results were processed and expressed by Thomas and Yoon-Nelson kinetic adsorption models.

Characterization of used pellets was determined by FTIR.

This manuscript is well written, all obtained results are clearly presented.

The research design could be improved by additional adsorption tests i.e. by batch sorption experiments and determination (at least) Langmuir and Freundlich Isotherm Studies.

Additional structural analysis (thermogravimetric analysis, nuclear magnetic resonance and scanning electron microscopy) of synthesized pellets would also improve the quality of this manuscript.

If the authors cannot provide additional informations related to structural analysis of tested adsorbents, the manuscript can be accepted in current form.

Response: While we agree with the reviewer that the use of batch adsorption results provide useful information on the adsorption properties, the kinetic results may not always show parallel differences (see Ref. [16]). Since this study was focused on the kinetic adsorption properties of the adsorbent materials and their practical application toward sulfate removal, the kinetic adsorption properties represent an independent approach that do not rely on batch adsorption results, as alluded to in the objectives in the Introduction section of the manuscript. Since the synthetic modification of the chitosan pellets was based on well-established synthetic methods, we limited the characterization to TGA and IR spectroscopic methods. These techniques provide a reliable indication of the types of functionalization and chemical modification of the chitosan framework, as reported elsewhere: see the following Refs: https://doi.org/10.1016/j.carbpol.2014.02.086; https://doi.org/10.1039/C5RA13981C; and Ref [25]) The approach outlined above provided complementary insight on the kinetic uptake properties of various chitosan materials toward sulfate in a fixed bed column system to meet the goals of this study, as outlined in the revised MS.

We acknowledge the insightful comments of Reviewer #1 and we have further edited the Manuscript for language, syntax and clarity throughout to meet the high standards of this journal.

Reviewer 2 Report

The manuscript entitled “Modified Biopolymer Adsorbents for Column Treatment of Sulfate Species in Saline Aquifers” with manuscript ID materials-808176 by Solgi et al. describes the sulfate uptake onto different forms of pelletized chitosan adsorbents using a fixed-bed column system.

The topic is interesting, the manuscript is well written and the results are clearly explained; therefore I suggest the manuscript publication in Materials with Minor Revisions based on the following remarks:

  1. In section 3.3 Table 1 should be mentioned in the text (around line 218) where the results from the table are mentioned.
  2. Why the authors choose pH values of 4.5 and 6.5 and no other pH values?
  3. The Figures are confusing since in black and white version is not possible to distinguish different data. I recommend using different types of markers together with the colors.
  4. In line 236 please rephrase “breakthrough model used to model adsorption”.
  5. The same unit type should be used along the text; for example qmax is expressed as mg g−1 in line 244 and mg/g in line 256.
  6. Equation 3 is not mentioned in the text.
  7. In section 3.5 what was the value of pH for the study of effect of operation conditions?
  8. In line 311 please correct “from 20 to 300 mm” with “from 200 to 300 mm”.
  9. In line 338 please rephrase to avoid the repetition “(Table 4). In Table 4…”
  10. In line 356 please correct 0.016 with 0.160.
  11. In table 5, the 6th column values represent Removal (%) or coefficient of determination (R2)? Please check and correct.
  12. Also please correct the values of R2 in line 376.
  13. In the “Desorption study” the authors mention that the decrease in the efficiency of adsorbent in cycles 3 and 4 is due to Ca leaching from adsorbent pellets; have the authors performed any analysis to prove this affirmation? An ICP analysis is recommended to support their affirmation.
  14. In line 408 the authors mention the sulfate uptake of chitosan in Table 7 with a qmax of 210 mg/g but I didn’t find this material in Table 7. Please rectify this error.
  15. The number of references in the text (32) doesn’t match with the number of references in the reference list (42).
  16. The reference format in the list is not the same for all the references: some include DOI and some don’t.
  17. The corresponding author Wilson L.D. has 6 self-citations which are not strictly related to the topic of this article; I recommend reducing the number of the self-citations.

Author Response

Response to Reviewer Comments on MS ID: materials-808176

Reviewer #2

The manuscript entitled “Modified Biopolymer Adsorbents for Column Treatment of Sulfate Species in Saline Aquifers” with manuscript ID materials-808176 by Solgi et al. describes the sulfate uptake onto different forms of pelletized chitosan adsorbents using a fixed-bed column system.

The topic is interesting, the manuscript is well written and the results are clearly explained; therefore I suggest the manuscript publication in Materials with Minor Revisions based on the following remarks:

  1. In section 3.3 Table 1 should be mentioned in the text (around line 218) where the results from the table are mentioned.

Response: “Table 1” has been added in the text in the revised MS.

  1. Why the authors choose pH values of 4.5 and 6.5 and no other pH values?

Response: According to ref. [20] the optimum pH for adsorption of sulfate on chitin was between 4 and 4.5. Also, ground water in many areas has pH around 6.5 to 7. So, in the present study pH 4.5 and 6.5 have been chosen to compare the result of adsorption. At pH 4.5 almost 99 % of the active sites on the surface of chitosan are protonated and pellets can adsorb more sulfate. This idea is also supported by adsorption results in this study. pH 6.5 has been chosen to investigate the application of new chitosan-based adsorbent for removal of sulfate from ground water.

  1. The Figures are confusing since in black and white version is not possible to distinguish different data. I recommend using different types of markers together with the colors.

Response: The authors prefer to use colors to distinguish the various systems for consistency throughout the manuscript. An annotation will be added to the figures to refer to the online version for color coding (see annotation in Figure 1 caption).

  1. In line 236 please rephrase “breakthrough model used to model adsorption”.

Response: The sentence has been corrected in the revised MS.

  1. The same unit type should be used along the text; for example, qmax is expressed as mg g−1 in line 244 and mg/g in line 256.

Response: Units have been corrected.

  1. Equation 3 is not mentioned in the text.

Response: It has been added to the revised MS.

  1. In section 3.5 what was the value of pH for the study of effect of operation conditions?

Response: The pH in section 3.5 was 4.5 and the value was added to the revised MS.

  1. In line 311 please correct “from 20 to 300 mm” with “from 200 to 300 mm”.

Response: The number has been corrected.

  1. In line 338 please rephrase to avoid the repetition “(Table 4). In Table 4…”

Response: Sentence has been rephrased in revised MS.

  1. In line 356 please correct 0.016 with 0.160.

Response: The value has been corrected.

  1. In table 5, the 6th column values represent Removal (%) or coefficient of determination (R2)? Please check and correct.

Response: The values are related to coefficient of determination and have been corrected in the table.

  1. Also please correct the values of R2 in line 376.

Response: According to the response for query #11 it does not need to change R2 in the revised MS.

  1. In the “Desorption study” the authors mention that the decrease in the efficiency of adsorbent in cycles 3 and 4 is due to Ca leaching from adsorbent pellets; have the authors performed any analysis to prove this affirmation? An ICP analysis is recommended to support their affirmation.

Response: According to the results in ref. [25], Ca2+ ions on the surface of chitosan beads were leached into the influent solution upon the adsorption process. It was reported that for cross-linked chitosan pellet after 1800 min, 1.2 ppm leached calcium was measured in the solution. Also, according ref. [25] and our results it is obvious that presence of calcium on the surface of adsorbent leads to increase the adsorption of sulfate by pellets. As a result, it can be inferred that decreasing of the maximum adsorption capacity of Ca-CP in cycle 3 and 4 is related to the leaching of Ca2+ during the adsorption-desorption process.

  1. In line 408 the authors mention the sulfate uptake of chitosan in Table 7 with a qmax of 210 mg/g but I didn’t find this material in Table 7. Please rectify this error.

Response: The value of 210 mg/g was wrong and has been removed accordingly.

  1. The number of references in the text (32) doesn’t match with the number of references in the reference list (42).

Response: References have been corrected.

  1. The reference format in the list is not the same for all the references: some include DOI and some don’t.

Response: DOI for references #13, 24,31 were added. There is no DOI available for references # 4, 5, 6, 22.

  1. The corresponding author Wilson L.D. has 6 self-citations which are not strictly related to the topic of this article; I recommend reducing the number of the self-citations.

Response: Ref. [14] has been added and ref. [23] has been exchange with a more relevant citation as recommended by the reviewer. There are 5 self-citations among the total of 44 citations listed in the revised MS.

The authors acknowledge the insightful comments of Reviewer #2 and we have further edited the Manuscript to reflect the above comments, along with editing of language, syntax and clarity throughout to meet the high standards of this journal.

Reviewer 3 Report

Generalities

The manuscript entitled “Modified biopolymers adsorbents for column treatment of sulfate species in saline aquifers” aims to modify different pelletized chitosan adsorbents to remove sulfate from saline aquifers. The manuscript needs of small modifications.

Specifications

  • Line 120. If the objective is test the suitability of the treatment with drinking water, why the system was tested at pH 4.5? Moreover, groundwater usually has a pH higher than 7. Why an slightly basic pH (i.e. 7.5 – 8.0 pH) was not tested?
  • Apparently, the different materials work better with acidic pH. In a real application, does the cost needed to decrease the pH compensate the higher removals observed?
  • How would the presence of other ions commonly found in groundwater (e.g. nitrate, chloride, calcium, etc.) affect the results obtained?

Author Response

Response to Reviewer Comments on MS ID: materials-808176

Reviewer #3

Generalities

The manuscript entitled “Modified biopolymers adsorbents for column treatment of sulfate species in saline aquifers” aims to modify different pelletized chitosan adsorbents to remove sulfate from saline aquifers. The manuscript needs of small modifications.

Specifications

  • Line 120. If the objective is test the suitability of the treatment with drinking water, why the system was tested at pH 4.5? Moreover, groundwater usually has a pH higher than 7. Why a slightly basic pH (i.e. 7.5 – 8.0 pH) was not tested?

Response: According to ref. [20] the optimum pH for adsorption of sulfate on chitin was between 4 and 4.5. also, ground water in many areas has pH around 6.5 to 7. So, in the present study pH 4.5 and 6.5 were chosen to compare the result of adsorption. At pH 4.5 ca. 99 % of the active sites on the surface of chitosan are protonated and pellets can adsorb more sulfate. This idea is supported by the adsorption results herein. pH 6.5 was chosen to investigate the application of new chitosan-based adsorbent for removal of sulfate from ground water.

  • Apparently, the different materials work better with acidic pH. In a real application, does the cost needed to decrease the pH compensate the higher removals observed?

Response: According to the facile method for preparing the chitosan pellets herein, the relative cost of these materials are estimated to be lower than the other type adsorbents in Table 7. Also, metal imbibing modification with calcium ions does not need any additional input of energy which is cost effective in practical applications. Although the maximum adsorption capacity was obtained at pH 4.5, adsorption result showed that in each cycle at pH 6.5 which is close to pH of various groundwater sources, all forms of adsorbents can remove more than 50 % of sulfate of influent solution. Thus, it can be concluded that these adsorbents may serve as promising materials for sulfate removal under practical conditions.

  • How would the presence of other ions commonly found in groundwater (e.g. nitrate, chloride, calcium, etc.) affect the results obtained?

Response: We agree with the reviewer that the presence of other anions in the influent solution may result in a decrease in the adsorption performance of the systems reported herein. This is understood on the basis of an ion-exchange mechanism and the role of pH as reported in this study. While the role of competitor ion adsorption is outside the objectives of the present study, this topic will be explored as part of a future planned study. We thank the reviewer for this valuable suggestion.

The authors acknowledge the insightful comments of Reviewer #3 and we have further edited the Manuscript to reflect the above comments, along with editing of language, syntax and clarity throughout to meet the high standards of this journal.
